# Nurses’ Perceptions of Electronic Medical Record Effectiveness at Ministry of Health Hospitals in Jeddah City: A Cross-Sectional Study

**DOI:** 10.3390/nursrep15090329

**Published:** 2025-09-09

**Authors:** Ebtihal Abdullah Rajab, Sabah Mahmoud Mahran, Nabeela Al Abdullah

**Affiliations:** 1Jeddah First Health Cluster, East Jeddah General Hospital, Jeddah 22253, Saudi Arabia; 2Public Health Department, King Abdulaziz University, Jeddah 21589, Saudi Arabia; smahran@kau.edu.sa (S.M.M.);

**Keywords:** electronic medical records, effectiveness, healthcare, nurses, system quality, user satisfaction

## Abstract

**Background:** Globally, there is a growing demand for the adoption of electronic health systems and the transition toward digital processes within healthcare organizations. Electronic Medical Records (EMRs) play a vital role in enhancing documentation accuracy, improving healthcare delivery, and minimizing medical errors. However, limited research has explored nurses’ perceptions of EMR effectiveness within Ministry of Health hospitals in Jeddah City. **Methods**: A quantitative descriptive cross-sectional design was employed in four governmental hospitals affiliated with the Ministry of Health in Jeddah. A convenience sampling technique was used to recruit 911 full-time registered nurses from inpatient and outpatient departments. Data was collected through an electronic self-administered questionnaire evaluating EMR use, system quality, and user satisfaction. Descriptive and inferential statistical analyses were conducted using SPSS version 26. **Results**: The global EMR score (82%) reflected a high level of acceptance and integration of EMR systems among the nurses surveyed. The use of order entry received the highest mean score (84.8%), indicating that nurses find EMRs particularly effective in streamlining administrative and clinical tasks, such as medication orders and care plans. The strong correlation between system quality and user satisfaction (rs = 0.911) underscores the importance of well-designed EMRs in fostering trust and confidence among clinical users. **Conclusions**: The findings indicate that nurses perceive EMRs as effective tools for improving documentation, care coordination, and workflow efficiency. This study recommends the establishment of structured feedback mechanisms that enable nurses to report issues, suggest improvements, and share success stories—thereby fostering a culture of continuous system enhancement.

## 1. Introduction

Over the past few decades, technological advancements have profoundly reshaped various domains of life, including communication, education, industry, and, notably, healthcare. Emerging innovations—such as artificial intelligence, big data analytics, robotics, and digital health systems—have significantly transformed societal operations and expectations. This digital evolution has intensified the demand for healthcare institutions to adopt modern technologies and digital solutions [1].

Electronic Medical Records (EMRs) serve as digital platforms that enable healthcare providers to access and manage patient information efficiently and securely. These systems support a variety of functions, including appointment scheduling, laboratory and imaging result reviews, order entry, and real-time documentation updates. Their implementation contributes to enhancing documentation accuracy and improving healthcare delivery while minimizing the likelihood of medical errors due to miscommunication [2].

A study conducted by Lloyd [3] in Australia aimed to explore the perspectives of medical and nursing clinicians on EMR usability. With a sample of 85 doctors and 27 nurses, the study highlighted positive aspects such as the ability to obtain diagnostic test results, ease of documenting medications, and access to information from any location. However, usability issues were also reported, including a lack of intuitiveness, system complexity, communication challenges with primary and relevant care sectors, and the time-consuming nature of completing clinical tasks. This study reflects global challenges in EMR implementation, offering useful comparisons to regional experiences like those in Saudi Arabia.

As a result, nurses are expected to continuously enhance their digital competencies to utilize EMRs effectively. The primary aim of EMR implementation is to improve healthcare quality by reducing clinical errors, fostering clear communication, enabling data sharing among professionals, and facilitating data collection for research and education [4]. However, this transition is not without obstacles. Some healthcare professionals initially resisted EMR adoption, citing challenges such as inadequate digital literacy, poor customization of systems to clinical needs, and insufficient IT support and training [5].

In the Saudi context, a study by Alzghaibi [6] emphasized that financial investment plays a pivotal role in the successful implementation of electronic health record systems. Additionally, research conducted by Jabber [7] explored nurses’ views on EMR usability, system quality, and satisfaction. Another relevant study by Alharbi [8] employed a quantitative cross-sectional survey with a sample of 521 healthcare providers across Saudi Arabia. It aimed to assess their knowledge and use of EMRs. The results indicated high levels of EMR understanding among both Saudi and non-Saudi participants but also highlighted potential deficiencies in practical usage due to limited training.

Furthermore, a nationwide study by Al-Otaybi [9] utilized a cross-national design involving 2684 healthcare workers to explore their perceptions and practices related to implemented EMR systems in Saudi Arabia. The findings revealed a generally satisfactory level of acceptance among participants—particularly senior staff and non-Saudi employees who had received EMR-related training. Despite multiple studies in Riyadh and other regions, limited research has addressed the specific context of Jeddah. The city’s unique healthcare infrastructure, population dynamics, and Ministry of Health facility distribution warrant focused investigation.

Addressing this geographic gap is crucial to understanding the challenges and opportunities faced by nurses in this locality. Despite ongoing advancements, nurses continue to encounter significant barriers when using EMRs, including system inefficiencies, dissatisfaction with platform design, and limited support channels. These issues can lead to increased stress and hinder integration into daily clinical workflows [10]. The success of EMR adoption depends on a combination of factors, including user participation, leadership support, adequate training, and stable financial resources. Without these components, the effective integration of EMRs may be compromised.

### Aim of the Study

This study aimed to evaluate nurses’ perceptions regarding the effectiveness of Electronic Medical Records (EMRs) in Ministry of Health hospitals in Jeddah City and to assess how these perceptions influence the overall system performance within healthcare centers.

## 2. Methodology

### 2.1. Research Design

This research utilized a quantitative descriptive cross-sectional design.

This approach is widely used in healthcare research to obtain a snapshot of current perceptions and behaviors across a defined population at a specific point in time.

### 2.2. Research Setting

This study was conducted at four hospitals under the Ministry of Health in Jeddah.

First is East Jeddah Hospital, a government hospital with 300 beds and a total of 791 staff nurses. This hospital is considered a trauma center. It offers a variety of medical specializations and has gained Saudi Central Board for Accreditation of Healthcare Institutions (CBAHI) accreditation. It was established in 2016.

Second is King Fahd Hospital. This governmental hospital was established in 1980 and is considered the largest healthcare facility of the Ministry of Health in Jeddah. The hospital provides several specialized centers and departments, including centers for heart surgery, laparoscopic surgery, and nuclear medicine. It has a capacity of 600 beds and 1062 staff nurses. The hospital has received accreditation from the Saudi Central Board for Accreditation of Healthcare Institutions (CBAHI).

Third, is King Abdullah Medical Complex. This governmental hospital opened in 2013 with a 500-bed capacity under the Ministry of Health. It provides elective medical and surgical services combined with a state-of-the-art laboratory, trauma center, and outpatient clinics serving North Jeddah. It employs a total of 987 staff nurses. The hospital received accreditation from the International Accreditation Commission (JCI) and the Saudi Center for Accreditation of Healthcare Institutions (CBAHI).

Finally, King Abdulaziz Hospital (Al-Mahjar), is a governmental hospital constructed in 1990. It has 445 beds available for numerous medical specialties, with a total of 780 staff nurses. The hospital has received accreditation from the Saudi Central Board for Accreditation of Healthcare Institutions (CBAHI).

All four hospitals provide non-profit, free healthcare services to the public and are accredited by the Central Board for Accreditation of Healthcare Institutions (CBAHI).

### 2.3. Sampling and Sample Size

The research utilized a convenience sampling method. The target population included 3620 registered nurses working full-time in inpatient and outpatient departments across the four selected hospitals. The required sample size was calculated using the Raosoft sample size calculator, with a 50% response distribution, a 95% confidence level, and a 3% margin of error. This yielded a minimum required sample of 825 participants. To ensure sufficient statistical power and to account for potential nonresponse or incomplete surveys, the sample size was increased by 10%, resulting in a final target of 911 participants.

Inclusion criteria were as follows: registered nurses who worked full-time in inpatient and outpatient departments and used electronic health records for daily documentation.

Exclusion criteria were as follows: intern nurses, nurse managers, and nurse educators who did not regularly utilize the EMR system as a component of their daily practice.

### 2.4. Data Collection Tools

The data collection was carried out through an online questionnaire. The first section of the questionnaire collected socio-demographic data, including gender, age, education level, nationality, department, work experience, and years of using EMRs. The second section, developed by [11], aimed to evaluate the effectiveness of the EMR system. It comprised a 34-item questionnaire designed to measure three dimensions: use of EMRs, quality of EMR systems, and user satisfaction. Twelve questions assessed the “use of EMR,” defined as the frequency with which nurses used the system to complete patient care-related tasks. Twelve questions evaluated “quality of EMR,” defined as the assessment of the responsiveness and quality of EMR systems. Eight questions assessed “user satisfaction with EMR,” defined as the extent to which nurses perceived the EMR system as a vital tool for enhancing their job performance. One item was included as a global measure: “Overall, are you pleased with the EMR system at your hospital?”

A Likert scale ranging from 1 to 5 was used, where 1 = never/almost never and 5 = always/almost always. For the user satisfaction dimension, a Likert scale was applied from 1 = not at all to 5 = very great. Each question was scored using the Likert scale. Positive and negative perceptions among nurses in each domain were charted, with scores of “4” and “5” classified as positive and “1” and “2” as negative. Higher overall scores in the questionnaire and its domains indicated a more positive perception of EMR effectiveness.

### 2.5. Pilot Study

A pilot study was conducted with 30 nurses, representing approximately 10% of the total sample, selected from East Jeddah Hospital. The purpose was to evaluate the simplicity, clarity, and completion time of the questionnaire. Based on pilot feedback, no modifications were necessary. Nurses reported that the questionnaire was easy to read and understand, requiring approximately 4–6 min to complete. Participants involved in the pilot study were excluded from the final research sample.

### 2.6. Validity and Reliability of the Questionnaire

The questionnaire was reviewed by a panel of expert faculty members from the Nursing Administration and Research Department at King Abdulaziz University along with expert personnel from East Jeddah Hospital to assess its consistency and provide suggestions for improvement. Internal consistency was measured using Cronbach’s alpha, yielding a value of 0.9478, indicating a high level of internal reliability.

### 2.7. Data Collection Procedure

After obtaining approval from the Nursing Faculty at King Abdulaziz University and the Ethics Committee of the Ministry of Health, data were collected using an electronic self-report questionnaire. Nurses who met the inclusion criteria and agreed to participate were invited via email to complete the survey. The invitation letter explained this study’s objectives, emphasized the voluntary nature of participation, and assured confidentiality and data protection by the research team. Participants were required to provide informed consent prior to completing the survey. Reminders were sent daily to nonrespondents to improve the response rate. Upon completion, all responses were securely saved and used solely for research purposes, ensuring data integrity and confidentiality.

The period of data collection commenced on 18 January 2024 and concluded in March 2024. A total of 911 nurses participated by providing replies. This study received ethical approval from the Ministry of Health and King Abdulaziz University (Approval Code: A01754).

Prior to participation, all individuals were provided with a clear explanation of this study’s purpose, its procedures, and their rights as participants. Informed consent was obtained from each participant before they proceeded to complete the survey. This process ensured that participation was voluntary and that individuals had the opportunity to decline or withdraw at any stage without any negative consequences. Confidentiality and anonymity were also maintained throughout this study to protect participants’ privacy and personal information, in alignment with ethical research standards.

### 2.8. Data Analysis

Data was presented using numbers and percentages for categorical variables and means, standard deviations, and medians (min–max) for continuous variables. Associations between global EMR scores and socio-demographic characteristics were analyzed using the Mann–Whitney Z-test and Kruskal–Wallis H-test. Post hoc analysis was performed using the Dunn–Bonferroni test. Correlations between EMR domains were assessed using Spearman correlation analysis. Normality was tested using the Kolmogorov–Smirnov test, revealing that global EMR and domain scores followed a non-normal distribution, thus necessitating the use of non-parametric tests.

The use of non-parametric tests was justified by the results of the Kolmogorov–Smirnov test, which indicated a non-normal distribution of the data. Therefore, the Mann–Whitney Z-test and Kruskal–Wallis H-test were chosen for group comparisons due to their suitability for ordinal data without normality assumptions. The Spearman correlation was used to assess the strength and direction of associations between ranked variables. The Dunn–Bonferroni post hoc test was employed to manage multiple comparisons while maintaining statistical rigor.

Statistical significance was considered at *p* < 0.05. All analyses were conducted using the Statistical Package for the Social Sciences (SPSS) version 26 (IBM Corporation, Armonk, NY, USA).

## 3. Results

A total of 911 nurses completed the survey, which corresponded exactly to the calculated sample size, resulting in a 100% response rate among the targeted participants. This high response rate was achieved through consistent follow-up and daily reminders sent to eligible nurses. Table 1 presents the socio-demographic characteristics of the participants. More than half of the respondents (58.1%) were aged between 31 and 40 years. The majority were female (88.4%), while males represented 11.6%. Regarding educational background, 65.6% of the participants held a bachelor’s degree. Approximately 60% of the respondents were married, and 48.4% had 10 years or more of work experience. The most common working unit was inpatient services (40.4%). In terms of EMR usage, 31.1% of the nurses had been using the system for 1 to 3 years, and 38.7% were affiliated with King Abdullah Medical Complex.

Table 2 presents the results of the EMR instrument tool. Regarding the nursing care management subscale, the top three positively rated items by nurses were “Enter daily nursing care notes” (82%), “Document physical assessment of patients” (81.1%), and “Review patients’ problems” (75.7%).

For the order entry subscale, the top three positive items were “Obtain results of tests and investigations” (84%), “Obtain the results from new tests or investigations” (83%), and “Obtain information on investigation or treatment procedures” (81.2%).

In terms of the information quality subscale, the highest-rated items were “How often is the system accurate?” (83.2%), “How often does the system provide up-to-date information?” (80.5%), and “How often is the information clear?” (80.1%).

Concerning the service quality subscale, the item “How often is the system subject to frequent system problems and crashes?” achieved a relatively high positive rating (75.6%).

Finally, for user satisfaction, the top three positive items were “Do you feel the safety of patients has improved due to Electronic Medical Records?” (83.4%), “Do you feel your performance has improved due to Electronic Medical Records?” (80.2%), and “Overall, are you satisfied with the Electronic Medical Record system?” (78.6%).

The descriptive statistics of the EMR instrument and its subscales are presented in Table 3. The highest mean percentage score was recorded for the “order entry” subscale (84.8%), followed by the “information quality” subscale (82%) and the “nursing care management” subscale (81.8%), whereas the “service quality” subscale had the lowest mean score (76.4%). Regarding the main EMR domains, the mean percentage scores were 83% for “use of EMR,” 81.2% for “quality of EMR,” and 82% for “user satisfaction.” The overall mean percentage score across all EMR domains was 82%.

Figure 1, Figure 2 and Figure 3 illustrate the correlations among the EMR instrument domains. A strong, positive, and statistically significant correlation was observed between the use of the EMR system and the perceived quality of the system (rs = 0.733, *p* < 0.001), as shown in Figure 1. Similarly, a positive and significant correlation was found between the use of the EMR system and user satisfaction (rs = 0.578, *p* < 0.001), as depicted in Figure 2. Furthermore, the perceived quality of the EMR system demonstrated a very strong, positive, and statistically significant correlation with user satisfaction (r = 0.911, *p* < 0.001), as presented in Figure 3.

Figure 1 depicts the correlation between EMR use and the quality of EMR scores, showing a positive, highly statistically significant correlation (rs = 0.733; *p* < 0.001), suggesting that the increase in EMR use correlates with the increase in the quality of EMR.

Figure 2 shows a positive, highly statistically significant correlation between EMR use and user satisfaction scores (rs = 0.578; *p* < 0.001), indicating that whenever the score of EMR use increases, the user satisfaction score will also likely increase.

Figure 3 illustrates a positive, highly statistically significant correlation between the quality of EMR and user satisfaction scores (rs = 0.911; *p* < 0.001), suggesting that increasing the quality of EMR scores is associated with increasing user satisfaction scores.

Exploring the differences in the score of the overall EMR in terms of the socio-demographic characteristics of the nurses found that higher global EMR scores were associated with being a non-Saudis (Z = 4.230; *p* < 0.001), having fewer years of experience (Z = 2.390; *p* = 0.017), and working at inpatient services (H = 12.759; *p* = 0.005). No significant differences were observed between the overall EMR score in relation to age, gender, education, and the duration of EMR use (*p* > 0.05) (Table 4).

Exploring the differences in the score of the overall EMR in terms of the socio-demographic characteristics of the nurses found that higher global EMR scores were associated with being a non-Saudi (Z = 4.230; *p* < 0.001), having fewer years of experience (Z = 2.390; *p* = 0.017), working at inpatient services (H = 12.759; *p* = 0.005), and using EMR systems at King Abdullah Medical Complex (H = 29.642; *p* < 0.00). No significant differences were observed between the global EMR score in relation to age, gender, education. and the duration of EMR use (*p* > 0.05) (Table 5).

## 4. Discussion

This study assessed how nurses in Ministry of Health hospitals in Jeddah perceived the effectiveness of Electronic Medical Record (EMR) systems. The analysis of individual questionnaire items provided valuable insights into specific areas where nurses found EMRs particularly beneficial.

Within the nursing care management domain, tasks such as documenting physical assessments and entering daily nursing notes received high ratings (81.1% and 82%, respectively). These findings indicated that nurses were confident in using EMRs for routine clinical documentation, supporting previous research that demonstrated improved efficiency and accuracy in nursing records [2]. This aligns with findings from Alharbi [8], who observed that EMR utilization was notably high among experienced users, particularly those who had received proper training, thus reinforcing the need for structured training programs. Furthermore, Ramoo [12] found a significant association between EMR training and confidence levels, echoing the need for continuous professional development.

Similarly, in the domain of order entry, the highest scores were observed for retrieving test results (84%) and accessing procedural data (81.2%). These results emphasized the role of EMRs in enhancing clinical decision-making by providing timely access to essential patient information. Comparable findings were reported by Alanzi and Addo [13,14], who noted that faster access to diagnostic information improved workflow coordination and reduced treatment delays. Furthermore, differences among hospitals in this domain—especially the higher perceived utility in King Abdullah Medical Complex—may reflect disparities in infrastructure, system updates, or IT support availability.

Regarding information quality, participants rated the system’s accuracy (83.2%) and clarity of information (80.1%) positively. These outcomes aligned with the conclusions of Alsulame [15], who stressed the significance of precise and real-time data in improving clinical safety and operational efficiency. Reliable information plays a critical role in minimizing clinical errors and ensuring continuity of care. Jaber [7] also emphasized that nurses who perceived EMRs as accurate and accessible were more likely to integrate them into daily practice.

Although most EMR domains received strong evaluations, the service quality domain scored comparatively lower, primarily due to occasional technical issues and system crashes (75.6%). While these technical challenges did not critically impair overall usability, they highlighted the need for improvements in system reliability, performance, and technical support [5]. It is worth noting that older nurses and those working in smaller hospitals expressed relatively lower satisfaction in this domain, suggesting that targeted system improvements and support mechanisms could reduce usage-related stress and increase adoption among these subgroups. This finding is consistent with Attafuah [16], who identified that low system responsiveness and lack of IT support were key contributors to dissatisfaction among nurses in similar settings.

In the user satisfaction domain, nurses expressed strong agreement that EMRs contributed to enhanced patient safety (83.4%) and improved job performance (80.2%). Overall satisfaction with the EMR system was reported at 78.6%, reinforcing the positive influence of EMR integration on clinical practice. These findings are consistent with Jamal [17], who found that positive EMR experiences correlate with greater perceptions of safety and professional efficiency. This finding is particularly promising in the context of Saudi Arabia’s Vision 2030, which emphasizes digital transformation as a means to enhance healthcare quality and operational efficiency.

Additionally, cross-demographic comparisons revealed that female nurses and non-Saudis exhibited slightly more favorable perceptions across most EMR domains, which may reflect differences in training background, adaptation pace, or expectations. These variations highlight the importance of inclusive training strategies that consider demographic diversity. Alotaybi [9] similarly found that EMR acceptance was significantly higher among non-Saudi healthcare workers, especially those with prior EMR exposure and seniority.

### 4.1. Limitations and Strengths

This study employed a cross-sectional design with convenience sampling, which may introduce bias and limit generalizability beyond the selected hospitals. Despite the large sample size and high response rate, the findings reflect nurses’ perceptions at a specific point in time and may not capture longitudinal changes. Moreover, self-reported data can be influenced by individual biases or recall errors. Nevertheless, this study’s strengths lie in its focus on four major public hospitals in Jeddah, its high participation rate, and the use of a validated EMR effectiveness scale, which collectively enhance the reliability of its conclusions. Future investigations using mixed methods or longitudinal approaches could offer deeper insights into behavioral and systemic factors affecting EMR usage.

### 4.2. Recommendations

Based on the findings, healthcare policymakers should prioritize EMR infrastructure upgrades—especially in underperforming hospitals—to address technical barriers. Training programs tailored by age, experience, and nationality could further support consistent usage and satisfaction across diverse staff. It is also recommended that future research adopts longitudinal or experimental designs to assess the impact of system improvements over time. Finally, encouraging feedback loops between nurses and IT teams could enhance user experience and trust in digital solutions.

### 4.3. Strategic Implications

In summary, the favorable perceptions reported across key EMR domains reflect a positive trend toward digital health maturity in Saudi Arabia’s public sector. These results not only support the continuity of EMR implementation but also encourage further optimization to align with national goals under Vision 2030.

## 5. Conclusions

This research investigated nurses’ perceptions of the effectiveness of Electronic Medical Record (EMR) systems in four Ministry of Health hospitals located in Jeddah. The results revealed a generally favorable perspective among participants regarding EMR utilization, system quality, and user satisfaction. Nurses recognized EMRs as valuable tools that enhanced documentation practices, facilitated care coordination, and improved clinical workflows. However, certain challenges—particularly related to system performance and technical functionality—were identified and should be addressed to maximize the benefits of EMRs.

Additionally, the findings highlighted meaningful differences in perceptions across hospitals, nationalities, and clinical departments, reinforcing the role of organizational infrastructure and staff characteristics in shaping EMR experiences.

This study contributes locally by addressing a gap in Jeddah’s healthcare context, which has been underrepresented in previous Saudi EMR research. Internationally, it offers insights relevant to countries implementing large-scale EMR systems, particularly in regions with diverse healthcare professionals. This aligns with global efforts to digitize healthcare while tailoring systems to user needs.

From a practical standpoint, this study reinforces the need for customized EMR training programs, especially for older nurses and non-Saudi staff. It also emphasizes the value of upgrading EMR platforms in underperforming hospitals to ensure consistency in system responsiveness and support. As Saudi Arabia advances its Vision 2030 goals, optimizing EMR usability will be critical for achieving national healthcare excellence.

In conclusion, nurses in this study demonstrated a strong awareness of EMR benefits but also expressed valid concerns regarding service quality. Addressing these concerns through systemic improvements, inclusive feedback mechanisms, and sustained digital investment will be vital for enhancing EMR engagement and the long-term impact in the healthcare sector.

## Figures and Tables

**Figure 1 nursrep-15-00329-f001:**
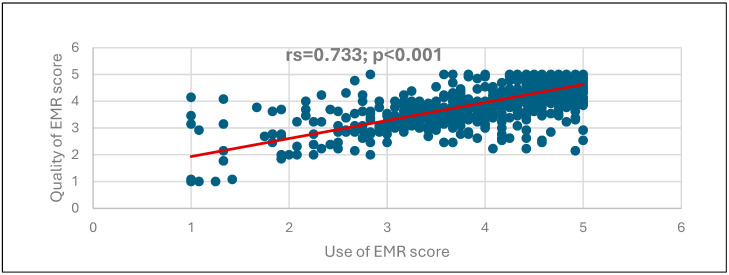
Correlation between the EMR use and the quality of EMR scores.

**Figure 2 nursrep-15-00329-f002:**
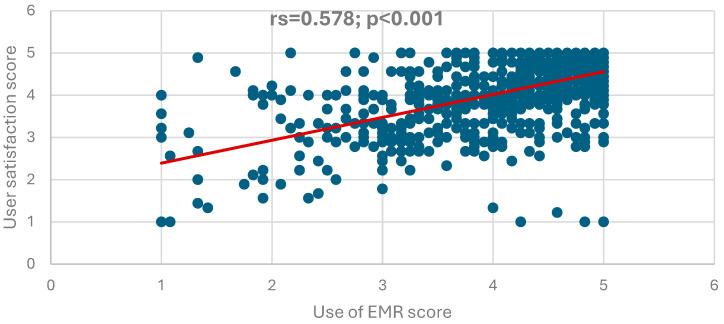
Correlation between the EMR use and the satisfaction of EMR scores.

**Figure 3 nursrep-15-00329-f003:**
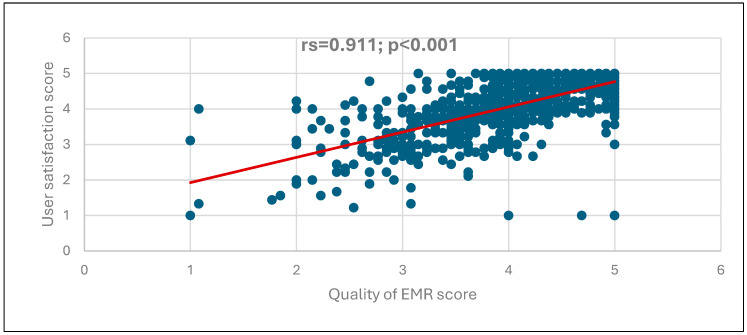
Correlation between the quality of EMR and user satisfaction scores.

**Table 1 nursrep-15-00329-t001:** Socio-demographic characteristics of the participating nurses (*n* = 911).

Study Variables	N = 911	(%)
Age group		
20–30 years	226	24.8%
31–40 years	529	58.1%
41–50 years	130	14.3%
>50 years	26	2.9%
Gender		
Male	106	11.6%
Female	805	88.4%
Nationality		
Saudi	708	77.7%
Non-Saudi	203	22.3%
Educational level		
Diploma holder	190	20.9%
Bachelor’s degree	598	65.6%
Postgraduate	123	13.5%
Marital status		
Single	364	40.0%
Married	547	60.0%
Work experience		
1–3 years	174	19.1%
4–6 years	162	17.8%
7–9 years	134	14.7%
≥10 years	441	48.4%
Working unit		
Critical care	195	21.4%
Inpatient services	368	40.4%
Outpatient services	183	20.1%
Operating room	50	5.5%
Emergency room	115	12.6%
How many years have you been using the EMR system?		
1–3 years	283	31.1%
4–6 years	282	31.0%
7–9 years	164	18.0%
10–12 years	89	9.8%
>12 years	93	10.2%
Name of the hospital		
East Jeddah Hospital	223	24.5%
King Abdulaziz Hospital	163	17.9%
King Abdullah Medical Complex	353	38.7%
King Fahad Hospital	172	18.9%

**Table 2 nursrep-15-00329-t002:** Frequency distribution and descriptive statistics of EMR instrument responses as perceived by staff nurses (*n* = 911).

EMR Use	Negative N (%)	Positive N (%)	Mean ± SD
Nursing care management			
Review patients problems.	82 (09.0%)	690 (75.7%)	4.10 ± 1.07
2. Enter daily nursing care notes.	85 (09.3%)	747 (82.0%)	4.31 ± 1.11
3. Capturing patient observations at the bedside.	112 (12.3%)	687 (75.4%)	4.01 ± 1.21
4. Write nursing care plans.	118 (13.0%)	679 (74.5%)	4.05 ± 1.25
5. Write nurse care worksheets (kardex).	163 (17.9%)	609 (66.8%)	3.78 ± 1.35
6. Collect patients’ information for discharge reports.	106 (11.6%)	684 (75.1%)	4.07 ± 1.22
7. Document physical assessment of patients.	74 (08.1%)	739 (81.1%)	4.28 ± 1.11
Frequency of use of order entry			
Obtain information on investigation or treatment procedures.	56 (06.1%)	740 (81.2%)	4.29 ± 1.01
2. Obtain the results from new tests or investigations.	53 (05.8%)	756 (83.0%)	4.35 ± 0.99
3. Answer questions concerning general medical knowledge (concerning treatment, symptoms, complications, etc.).	76 (08.3%)	701 (76.9%)	4.14 ± 1.07
4. Obtain results of tests and investigations.	55 (06.0%)	765 (84.0%)	4.35 ± 0.99
5. Check drug information (such as allergies and interactions).	103 (11.3%)	694 (76.2%)	4.09 ± 1.17
Information quality subscale			
How often does the system provide the precise information you need?	57 (06.3%)	714 (78.4%)	4.09 ± 0.95
2. How often does the information content meet your needs?	48 (05.3%)	715 (78.5%)	4.11 ± 0.94
3. How often does the system provide reports that seem to be just exactly what you need?	51 (05.6%)	701 (76.9%)	4.07 ± 0.93
4. How often does the system provide sufficient information?	55 (06.0%)	705 (77.4%)	4.09 ± 0.95
5. How often is the system accurate?	38 (04.2%)	758 (83.2%)	4.18 ± 0.85
6. How often are you satisfied with the accuracy of the system?	60 (06.6%)	720 (79.0%)	4.09 ± 0.96
7. How often do you think the output is presented in a useful format?	54 (05.9%)	703 (77.2%)	4.08 ± 0.93
8. How often is the information clear?	45 (04.9%)	730 (80.1%)	4.15 ± 0.89
9. How often is the system user-friendly?	70 (07.7%)	683 (75.0%)	4.02 ± 1.01
10. How often do you get the information you need in time?	48 (05.3%)	714 (78.4%)	4.10 ± 0.92
11. How often does the system provide up-to-date information?	60 (06.6%)	733 (80.5%)	4.12 ± 0.95
Service quality subscale			
How often can you count on the system to be up and available?	62 (06.8%)	681 (74.8%)	3.99 ± 0.96
2. How often is the system subject to frequent system problems and crashes?	62 (06.8%)	689 (75.6%)	3.65 ± 1.15
User satisfaction			
Do you feel Electronic Medical Records are useful?	179 (19.6%)	548 (60.2%)	4.16 ± 0.92
2. Do you feel your performance has improved due to Electronic Medical Records?	46 (05.0%)	731 (80.2%)	4.06 ± 0.93
3. Do you feel the quality of your work has improved?	51 (05.6%)	710 (77.9%)	4.06 ± 0.95
4. Do you feel that an Electronic Medical Record is worth the time and effort required to use it?	55 (06.0%)	700 (76.8%)	4.04 ± 0.95
5. Do you feel the quality of information has improved due to Electronic Medical Records?	61 (06.7%)	690 (75.7%)	4.09 ± 0.91
6. Do you feel Electronic Medical Records have been successful in your hospital?	51 (05.6%)	712 (78.2%)	4.08 ± 0.89
7. Do you feel that Electronic Medical Records are an important system in your hospital?	44 (04.8%)	709 (77.8%)	4.23 ± 0.90
8. Do you feel the safety of patients has improved due to Electronic Medical Records?	44 (04.8%)	760 (83.4%)	4.09 ± 0.92
9. Overall, are you satisfied With the Electronic Medical Record system?	62 (06.8%)	716 (78.6%)	4.09 ± 0.95

**Table 3 nursrep-15-00329-t003:** Overall scores of EMR instrument and its subscales (n = 911).

Variables	Mean ± SD	Mean (%)
Use of EMR score	4.15 ± 0.86	83.0%
Nursing care management subscale score	4.09 ± 0.93	81.8%
Use of order entry subscale score	4.24 ± 0.88	84.8%
Quality of EMR score	4.06 ± 0.78	81.2%
Information quality subscale score	4.10 ± 0.79	82.0%
Service quality subscale score	3.82 ± 0.92	76.4%
User satisfaction score	4.10 ± 0.79	82.0%
Overall EMR score	4.10 ± 0.72	82.0%

**Table 4 nursrep-15-00329-t004:** Association between global EMR score and the socio-demographic characteristics of the nurses (n = 911).

Factor	Overall EMR Score (5) Mean ± SD	Z/H-Test	*p*-Value
Age group ^a^			
20–30 years	4.16 ± 0.69	2.387	0.303
31–40 years	4.09 ± 0.73		
>40 years	4.05 ± 0.74		
Gender ^b^			
Male	4.07 ± 0.77	0.329	0.742
Gender	4.11 ± 0.72		
Nationality ^b^			
Saudi	4.05 ± 0.75	4.230	<0.001 **
Non-Saudi	4.29 ± 0.59		
Educational level ^a^			
Diploma holder	4.04 ± 0.74	2.975	0.226
Bachelor’s degree	4.12 ± 0.73		
Postgraduate	4.09 ± 0.63		
Marital status ^b^			
Single	4.05 ± 0.76	1.322	0.186
Married	4.13 ± 0.69		
Work experience ^b^			
<10 years	4.15 ± 0.72	2.390	0.017 **
≥10 years	4.05 ± 0.72		
Working unit ^a^			
Critical care	4.08 ± 0.72	12.759	0.005 **
Inpatient services	4.19 ± 0.68		
Outpatient services	3.96 ± 0.78		
OR/ER	4.09 ± 0.73		
Duration of using EMR system ^a^			
1–3 years	4.12 ± 0.75	1.428	0.490
4–6 years	4.11 ± 0.71		
>6 years	4.08 ± 0.71		

^a^ *p*-value has been calculated using Kruskal–Wallis H-test. ^b^ *p*-value has been calculated using Mann–Whitney Z-test. ** Significant at *p* < 0.05 level.

**Table 5 nursrep-15-00329-t005:** Association between EMR domain scores and the socio-demographic characteristics of the nurses (n = 911).

Factor	Use of EMR Score (5) Mean ± SD	Quality EMR Score (5) Mean ± SD	User Satisfaction Score (5) Mean ± SD
Age group ^a^			
20–30 years	4.25 ± 0.81	4.09 ± 0.79	4.16 ± 0.75
31–40 years	4.13 ± 0.85	4.04 ± 0.78	4.10 ± 0.80
>40 years	4.07 ± 0.94	4.07 ± 0.94	4.01 ± 0.85
H-test; *p*-value	3.392; 0.183	0.931; 0.628	2.560; 0.278
Gender ^b^			
Male	4.13 ± 0.85	4.06 ± 0.86	3.99 ± 0.91
Gender	4.15 ± 0.86	4.06 ± 0.77	4.11 ± 0.78
Z-test; *p*-value	0.295; 0.768	0.403; 0.687	1.007; 0.314
Nationality ^b^			
Saudi	4.09 ± 0.88	4.00 ± 0.81	4.05 ± 0.82
Non-Saudi	4.37 ± 0.75	4.26 ± 0.61	4.26 ± 0.69
Z-test; *p*-value	4.672; <0.001 **	3.583; <0.001 **	3.070; <0.001 **
Educational level ^a^			
Diploma holder	4.07 ± 0.91	4.05 ± 0.79	3.98 ± 0.77
Bachelor’s degree	4.17 ± 0.88	4.07 ± 0.79	4.14 ± 0.79
Postgraduate	4.17 ± 0.67	4.02 ± 0.72	4.11 ± 0.88
H-test; *p*-value	2.998; 0.223	8.750; 0.019 **	2.975; 0.226
Marital status ^b^			
Single	4.12 ± 0.89	3.99 ± 0.82	4.06 ± 0.83
Married	4.17 ± 0.83	4.11 ± 0.75	4.13 ± 0.78
Z-test; *p*-value	0.623; 0.533	2.082; 0.037 **	1.137; 0.256
Work experience			
<10 years	4.19 ± 0.86	4.10 ± 0.79	4.16 ± 0.76
≥10 years	4.10 ± 0.85	4.01 ± 0.76	4.04 ± 0.84
Z-test; *p*-value	2.136; 0.033 **	1.931; 0.054	1.949; 0.051
Working unit ^a^			
Critical care	4.15 ± 0.79	4.01 ± 0.77	4.11 ± 0.79
Inpatient services	4.29 ± 0.78	4.11 ± 0.76	4.17 ± 0.76
Outpatient services	3.92 ± 1.01	3.98 ± 0.81	3.97 ± 0.85
OR/ER	4.09 ± 0.86	4.08 ± 0.79	4.08 ± 0.82
H-test; *p*-value	20.943; <0.001 **	4.101; 0.251	6.904; 0.075
Duration of using EMR system ^a^			
1–3 years	4.21 ± 0.87	4.06 ± 0.80	4.10 ± 0.83
4–6 years	4.15 ± 0.85	4.05 ± 0.79	4.15 ± 0.77
>6 years	4.10 ± 0.86	4.06 ± 0.76	4.06 ± 0.79
H-test; *p*-value ^a^	3.392; 0.183	0.034; 0.983	2.270; 0.321
H-test; *p*-value	36.851; <0.001 **	11.283; 0.010 **	29.352; <0.001 **

^a^ *p*-value has been calculated using Kruskal–Wallis H-test. ^b^ *p*-value has been calculated using Mann–Whitney Z-test. ** Significant at *p* < 0.05 level.

## Data Availability

The data that support the findings of this study are available from the corresponding author upon reasonable request.

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
