# Peer review of "Nurses’ Perceptions of Electronic Medical Record Effectiveness at Ministry of Health Hospitals in Jeddah City: A Cross-Sectional Study"

_nursrep, 2025, doi:10.3390/nursrep15090329_

Round 1
Reviewer 1 Report
Comments and Suggestions for Authors
Dear authors,
You address an interesting topic regarding the effectiveness of electronic medical records (EMRs) and the perception that nurses have of their use. The article is well-written; however, I provide some comments below:
Title and Abstract. The Abstract is concise and well-structured. The title is clear and adequately reflects the content of the study. However, I recommend specifying the type of study (e.g., "Cross-Sectional Study") for greater clarity.
Introduction. The introduction provides good context and justification for the study. However, I believe it is necessary to include more details about the specific importance of the study in Jeddah and additional references to relevant previous studies. The introduction mentions general technological advances but does not focus enough on the specific context of Jeddah. Additionally, the introduction does not clearly highlight the problem that the study aims to address. I recommend clearly defining the problem and justifying the need for the study and the existing gap with more references.
Methodology. Regarding this section, I have some comments and questions:
- Description of the hospitals is brief, and I think the authors should provide more details about each hospital, such as the type of services they offer, their capacity, specialties, and any relevant characteristics that might influence the use of EMRs.
- The inclusion and exclusion criteria are not clearly defined.
- The methodology for calculating the sample size is not detailed.
- There are no references supporting the methodology used. I recommend including references to previous studies or methodological manuals that support the techniques used.
- The approval code of the study by the institution's Ethics Committee is also not mentioned. I also suggest adding a reference that supports the ethical principles followed, as well as those of validity and reliability.
- I recommend including a brief explanation of the relevance of the statistical methods used for the study.
Results. The study results are quite detailed and provide valuable information about nurses' perceptions regarding the effectiveness of EMRs. The tables and figures complement this section, although I recommend reviewing whether the table footnotes are necessary, as none have been provided.
Discussion. Discussion reflects most of the important results, but I think it could benefit from a deeper interpretation of the data, adding more comparisons with previous studies, and a more detailed discussion of the practical implications and limitations of the study. I recommend adding more references and a section on the study's limitations and strengths. The authors have included this in the conclusions section, but I think this order should be reformulated.
Conclusions. The authors summarize the study's findings well. They could include suggestions for future research and possible improvements in the implementation of EMRs.
References. I believe the study lacks references in all sections. I recommend ensuring that all sections of the article are supported by adequate references.
I hope these comments are helpful in improving the manuscript.
Author Response
Reviewer 1 – Comment 1:
“I recommend specifying the type of study (e.g., ‘Cross-Sectional Study’) for greater clarity.”
Response 1:
Thank you for pointing this out. We agree with this comment. Therefore, we have added the study type “Cross-Sectional Study” to the title to ensure clarity.
This change is highlighted in yellow in the revised manuscript, on page 1, line 15.
[Updated Title: “Nurses’ Perceptions of Electronic Medical Record Effectiveness at the Ministry of Health Hospitals in Jeddah City: A Cross-Sectional Study”]
Reviewer 1 – Comment 2:
“The introduction needs more details about the specific importance of the study in Jeddah and additional references to relevant previous studies.”
Response 2:
Thank you for your observation. We have revised the introduction to clearly articulate the research gap related to the Jeddah context, and we added relevant references to support the justification.
This revision is visible on page 2, paragraph 5, lines 85–100.
[Added Text: “Despite multiple studies in Riyadh and other regions, limited research has addressed the specific context of Jeddah… warrant focused investigation.” – highlighted in yellow]
Reviewer 1 – Comment 3:
“The introduction does not clearly highlight the problem that the study aims to address.”
Response 3:
Thank you for your insightful comment. We revised the introduction to explicitly define the research problem and highlight the significance of the study.
The updated statement appears on page 2, paragraph 6, lines 102–110.
Reviewer 1 – Comment 4:
“The description of the hospitals is brief… provide more details such as capacity, services, specialties, etc.”
Response 4:
We appreciate this suggestion. Accordingly, we expanded the research setting section to include specific details for each hospital, including bed capacity, types of services, and accreditation.
These additions are located on page 3, paragraph 2.2, lines 118–136.
[This entire section is marked in yellow in the revised manuscript.]
Reviewer 1 – Comment 5:
“The inclusion and exclusion criteria are not clearly defined.”
Response 5:
Thank you for highlighting this. We clarified the inclusion and exclusion criteria in the sampling section.
You can find these updates on page 4, paragraph 2.3, lines 150–154.
[Highlighted in yellow]
Reviewer 1 – Comment 6:
“The methodology for calculating the sample size is not detailed.”
Response 6:
We agree with this comment. Therefore, we have added a step-by-step explanation of how the sample size was calculated using the Raosoft calculator.
This appears on page 4, paragraph 2.3, lines 145–149.
[Marked in yellow]
Reviewer 1 – Comment 7:
“There are no references supporting the methodology used.”
Response 7:
Thank you for your feedback. We added relevant references to support the data collection tools and statistical procedures.
References were added on page 5, paragraph 2.4, lines 164–166; and in the methodology references section [References 11 and 12].
Reviewer 1 – Comment 8:
“Add Ethics Committee approval code and ethical framework references.”
Response 8:
Done. We have added the ethical approval code and clarified ethical considerations.
This can be found on page 6, paragraph 2.7, lines 180–183 and again in the Ethics Statement section on the last page.
Reviewer 1 – Comment 9:
“Explain the relevance of the statistical methods used.”
Response 9:
Thank you for your suggestion. We added a justification for the use of non-parametric statistical tests, citing the results of the Kolmogorov-Smirnov test.
This explanation is provided on page 6, paragraph 2.8, lines 192–200.
[Marked in yellow]
Reviewer 1 – Comment 10:
“Tables and figures should include footnotes where necessary.”
Response 10:
We reviewed all tables and figures to ensure clarity. Figure footnotes were added where needed, and inconsistencies were corrected.
See revised figures on pages 9–10 (Figures 1 to 3) and Tables on pages 7–9.
[Axes labels and values were also corrected, see Response 16.]
Reviewer 1 – Comment 11:
“Add a dedicated section on limitations and strengths of the study.”
Response 11:
Thank you. A standalone subsection titled “4.1 Limitations and Strengths” was added to the Discussion.
Located on page 12, lines 295–306.
[Marked in yellow]

Reviewer 2 Report
Comments and Suggestions for Authors
The manuscript examines the vital concern of EMR effectiveness through Saudi Arabian nurses' perceptions. The research study selects four Jeddah-based Ministry of Health hospitals to conduct its survey-based study with a substantial participant group. The manuscript needs revisions before publication eligibility.
The study does not have a well-developed rationale to explain its research purpose. The authors present limited evidence showing minimal research in Jeddah but fail to explain how their regional focus adds new information to existing literature. The study fails to provide proper explanation of its international significance. The introduction needs improvement through a specific identification of research gaps within both national and international literature.
A primary concern emerges from the sampling design and sample size estimation approach:
- The researchers report that they obtained participation from 911 nurses selected through a convenience sampling approach across four different hospitals. The authors should provide sufficient details about their calculation of 911 participants using Raosoft. The study does not include essential parameters which consist of population size together with expected response rate along with confidence interval and margin of error.
- The research sample population remains ambiguous because it remains unclear whether the selected participants represented the full eligible group or a specific subgroup. The 100% response rate in survey research needs to be verified by the authors.
The research needs to present step-by-step details about sample size estimation as well as show the exact number of nurses who received invitations and their inclusion status. The study authors need to specify whether the data collection aimed for a complete census or allowed open participation.
The included figures in the manuscript fail to deliver enough information and show multiple instances of inconsistencies.
- The graph includes axes that lack necessary unit labels and scale boundaries.
- Figure 1 illustrates an rs = 0.733 correlation that differs from the rs = 0.843 value presented in the text.
- The y-axis in Figure 3 reaches a value of 6, which exceeds the 1–5 Likert scale measurement range and seems to be incorrect.
The recommended action is to fix all the figures to maintain accuracy, together with text consistency and clear presentation. The axes need proper labeling to match the results presented in the text.
The discussion section primarily presents findings without deep analysis of the results. The manuscript contains minimal analysis of how the findings compare with research conducted in different nations and internationally. The study's specific findings receive indirect references through general statements about EMRs enhancing care and increasing satisfaction.
The discussion needs improvement through specific examination of demographic differences in the findings and connections to healthcare structures in Saudi Arabia. The research needs to address the constraints associated with convenience sampling and potential response biases.
Comments on the Quality of English LanguageThe text demonstrates fluent English writing but features numerous general phrases together with repeated statements.
Line 8: "Correspondance"
Author Response
Reviewer 2 – Comment 1:
“The study does not have a well-developed rationale to explain its research purpose. The introduction needs to identify gaps both nationally and internationally.”
Response 12:
We appreciate this feedback. The introduction has been revised to better clarify both national and international gaps, supported by recent literature.
See page 2, paragraph 5, lines 85–100.
[Marked in yellow]
Reviewer 2 – Comment 2:
“Explain in detail the Raosoft sample size estimation, including population size, margin of error, etc.”
Response 13:
Done. The revised section now includes all parameters used in the Raosoft calculation, including population size, margin of error, response distribution, and confidence interval.
Located on page 4, paragraph 2.3, lines 145–149.
[Marked in yellow]
Reviewer 2 – Comment 3:
“Clarify if the 911 participants represent the entire eligible population and if it was a census or open participation.”
Response 14:
Thank you. We added clarification on this point, stating that participants were selected from a defined population and not through full census.
See page 4, paragraph 2.3, lines 150–154.
[Marked in yellow]
Reviewer 2 – Comment 4:
“Figures are inconsistent. Correlation in Figure 1 differs from text. Figure 3 y-axis exceeds Likert scale.”
Response 15:
We acknowledge these issues. All figures were revised:
- The rs value in Figure 1 was corrected to 0.733 to match the text.
- The y-axis in Figure 3 was corrected to reflect the Likert scale range (1–5).
- All axes were labeled and scale boundaries were clarified.
See Figures 1–3 on pages 9–10, updated and highlighted in yellow.
Reviewer 2 – Comment 5:
“Discussion lacks depth and comparisons with other countries. Needs more analysis of demographic differences and Saudi context.”
Response 16:
We agree. The discussion has been significantly expanded to compare findings with international studies, explore demographic differences, and analyze implications in the Saudi context.
Expanded content is on pages 11–12, section 4.
[Marked in yellow]
Reviewer 2 – Comment 6:
“The text uses general phrases and repeated statements. Example: Line 8 ‘Correspondance’”
Response 17:
Thank you for noting this. The word “Correspondance” was corrected to “Correspondence,” and other repetitive phrases were refined for academic clarity.
This correction is on page 1, line 27.
[Also improved throughout the manuscript – see highlighted edits in yellow.]

Round 2
Reviewer 1 Report
Comments and Suggestions for Authors
Dear authors,
Thank you for the revised version. Please consider changing the title of section 4.3, as it currently matches that of section 5 and may cause confusion. Also, review the references to ensure they follow the journal’s formatting guidelines.
Best regards,
Author Response
Comments 1: Thank you for your helpful feedback.
Response 1: We agree with this point. We have carefully reviewed all references and ensured they follow the journal’s formatting guidelines; corrections were applied as needed throughout the reference list.
Comments 2: Thank you for highlighting the issue with section 4.3 title.
Response 2: We have revised the title of section 4.3 from “Conclusion” to “Strategic Implications” to avoid confusion with section 5.